Three-dimensional geometrical changes of the human tibialis anterior muscle and its central aponeurosis measured with three-dimensional ultrasound during isometric contractions

Raiteri Brent J. b.raiteri@uq.edu.au
Cresswell Andrew G.
Lichtwark Glen A.
Centre for Sensorimotor Performance, School of Human Movement and Nutrition Sciences, The University of Queensland , Brisbane , Queensland , Australia
Abdala Virginia
Electronic publication date: 2016 Jul 28
Publication date: 2016
Volume: 4
Electronic Location ID: e2260
Received 2016 Apr 26; Accepted 2016 Jun 25
Copyright: ©2016 Raiteri et al.
Copyright year: 2016
Copyright holder: Raiteri et al.
License: This is an open access article distributed under the terms of the Creative Commons Attribution License, which permits unrestricted use, distribution, reproduction and adaptation in any medium and for any purpose provided that it is properly attributed. For attribution, the original author(s), title, publication source (PeerJ) and either DOI or URL of the article must be cited.
License URL: https://creativecommons.org/licenses/by/4.0/

Keywords: Contraction intensity, Muscle force, Muscle bulging, Aponeuroses, Muscle architecture

Funding: Australian Postgraduate Award Infrastructure support came from The University of Queensland. Brent J. Raiteri is supported by an Australian Postgraduate Award. The funders had no role in study design, data collection and analysis, decision to publish, or preparation of the manuscript.

==============================
Background. Muscles not only shorten during contraction to perform mechanical work, but they also bulge radially because of the isovolumetric constraint on muscle fibres. Muscle bulging may have important implications for muscle performance, however quantifying three-dimensional (3D) muscle shape changes in human muscle is problematic because of difficulties with sustaining contractions for the duration of an in vivo scan. Although two-dimensional ultrasound imaging is useful for measuring local muscle deformations, assumptions must be made about global muscle shape changes, which could lead to errors in fully understanding the mechanical behaviour of muscle and its surrounding connective tissues, such as aponeurosis. Therefore, the aims of this investigation were (a) to determine the intra-session reliability of a novel 3D ultrasound (3DUS) imaging method for measuring in vivo human muscle and aponeurosis deformations and (b) to examine how contraction intensity influences in vivo human muscle and aponeurosis strains during isometric contractions.

Methods. Participants (n = 12) were seated in a reclined position with their left knee extended and ankle at 90° and performed isometric dorsiflexion contractions up to 50% of maximal voluntary contraction. 3DUS scans of the tibialis anterior (TA) muscle belly were performed during the contractions and at rest to assess muscle volume, muscle length, muscle cross-sectional area, muscle thickness and width, fascicle length and pennation angle, and central aponeurosis width and length. The 3DUS scan involved synchronous B-mode ultrasound imaging and 3D motion capture of the position and orientation of the ultrasound transducer, while successive cross-sectional slices were captured by sweeping the transducer along the muscle.

Results. 3DUS was shown to be highly reliable across measures of muscle volume, muscle length, fascicle length and central aponeurosis length (ICC ≥ 0.98, CV < 1%). The TA remained isovolumetric across contraction conditions and progressively shortened along its line of action as contraction intensity increased. This caused the muscle to bulge centrally, predominantly in thickness, while muscle fascicles shortened and pennation angle increased as a function of contraction intensity. This resulted in central aponeurosis strains in both the transverse and longitudinal directions increasing with contraction intensity.

Discussion. 3DUS is a reliable and viable method for quantifying multidirectional muscle and aponeurosis strains during isometric contractions within the same session. Contracting muscle fibres do work in directions along and orthogonal to the muscle’s line of action and central aponeurosis length and width appear to be a function of muscle fascicle shortening and transverse expansion of the muscle fibres, which is dependent on contraction intensity. How factors other than muscle force change the elastic mechanical behaviour of the aponeurosis requires further investigation.

Introduction

Muscle fibres remain isovolumetric during skeletal muscle contraction (Huxley, 1953; Elliott, Lowy & Worthington, 1963) and simultaneously shorten along their line of action and bulge radially during active force production (Wakeling & Randhawa, 2014). Muscle fibre shortening therefore creates transverse and longitudinal forces within the muscle during contraction and may stretch the elastic connective tissues, such as aponeurosis, bi-axially (Azizi & Roberts, 2009). Variations in the amount of fibre shortening and transverse expansion at different contraction forces may change the elastic behaviour of the aponeurosis, by altering its width and length. For instance, the transverse strain of the turkey lateral gastrocnemius aponeurosis has been shown to increase with muscle force up to 20% of maximum isometric force and subsequently increase the longitudinal stiffness of the aponeurosis by two to three times during isotonic contractions compared to passive stretches (Azizi & Roberts, 2009). This mechanism for varying series elasticity could have significant implications for our predictions of work output in muscles that rely heavily on elastic strain energy storage and return during stretch-shorten cycles (Wilson & Lichtwark, 2011).

Understanding how muscle bulging influences muscle behaviour in human muscles remains problematic, due to difficulties in performing steady submaximal contractions for the length of time it takes to image the entire muscle with in vivo techniques, such as magnetic resonance imaging. To our knowledge, only one investigation has determined in vivo human aponeurosis width and length changes concurrently during contraction. This ultrasound imaging study by Maganaris and colleagues (2001) found an increase in central aponeurosis width and length of the tibialis anterior (TA) muscle during maximal dorsiflexion contractions. However, the aponeurosis length increase (7%) was much greater than the length increase (1.6%) subsequently estimated in a study by Tilp and colleagues (2012), suggesting that that two-dimensional ultrasound (2DUS) imaging techniques may not be accurate for quantifying aponeurosis strains. Interestingly, the study by Tilp, Steib & Herzog (2012) also found that central aponeurosis length decreased at approximately 10% of maximal voluntary isometric contraction (MVIC), and although this finding could be due to measurement error, it may indicate that at low activations the shortening of muscle fibres produces a transversal rather than longitudinal aponeurosis strain and a subsequent reduction in aponeurosis length (Azizi & Roberts, 2009).

One potential method for more accurately determining both longitudinal and transverse strains of the aponeurosis during contractions is three-dimensional ultrasound (3DUS) imaging (Barber, Barrett & Lichtwark, 2009). This technique has been used to measure longitudinal and transverse strains of the proximal Achilles tendon during isometric plantar flexion contractions (Farris et al., 2013) and should also be suitable to provide more detailed quantification of how factors such as muscle shortening and bulging influence strains of connective tissue.

Figure 1 Schematic representations of the shank in the sagittal plane and the tibialis anterior muscle in the sagittal and transverse planes with its geometrical parameters defined.

(A) A sagittal plane representation of the shank with the tibialis anterior (TA), tibialis posterior (TP), soleus (SOL) and lateral gastrocnemius (LG) muscles identified. (B) A sagittal plane representation of the TA muscle showing its bi-pennate architecture with the superficial and deep muscle fascicles (oblique red lines) at an angle to the aponeuroses (horizontal blue lines) with the free tendon (horizontal black line) in-series with the central aponeurosis. Measures of central aponeurosis length, fascicle length, pennation angle and muscle length have been defined in the sagittal plane. (C) A transverse plane representation of the TA muscle showing its central aponeurosis (horizontal blue line) separating the superficial and deep muscle compartments. The ultrasound transducer in (A) identifies the slice location. Measures of central aponeurosis width, muscle thickness and muscle width have been defined in (C) and the shaded salmon area shows muscle cross-sectional area (CSA).

The aim of this study was to use a novel freehand, 3DUS imaging technique to measure the in vivo human TA muscle and aponeurosis deformations (defined in Fig. 1) during submaximal isometric dorsiflexion contractions from rest to 50% MVIC. We first determined the intra-session reliability of using 3DUS to make measures of muscle volume, muscle length, and aponeurosis width and length at rest and during each submaximal contraction intensity. After assessing the reliability, we tested the hypothesis that the muscle fascicles and muscle belly would shorten while the muscle cross-sectional area (CSA) would increase as a function of contraction intensity, so that muscle volume would remain relatively constant. We further hypothesized that bulging of the muscle, orthogonal to its line of action, would be driven by progressive increases in muscle thickness from increases in fascicle pennation angle for the low to moderate forces.

Methods

Participants

Twelve participants (age: 24 (mean) ± 2 (standard deviation) years, height: 175 ± 7 cm, body mass: 69 ± 10 kg, 6 males) with no recent (<12 months) history of lower limb injury or surgery and no preexisting neuromuscular disorders, volunteered to participate in the study. All participants were recreationally active at the time of testing and provided written informed consent prior to participation. The study protocol was conducted in accordance with the Declaration of Helsinki and approved (HMS14/0704) and endorsed by the University of Queensland Human Research Ethics Committee (HREC).

Experimental setup and design

Participants were seated in a reclined position with their hips at 130° and the plantar aspect of their left foot flush against a custom-built rigid footplate (Fig. 2). The left knee was fully extended and the left ankle was at 90° (base of foot approximately perpendicular to shank). Padded wooden blocks were positioned between the superior dorsal aspect of the left foot and the frame housing the foot to minimize ankle rotation during dorsiflexion contractions.

Figure 2 Experimental setup.

A schematic of the footplate and load cell setup and the participant’s position during the three-dimensional (3D) ultrasound scans, which combined B-mode imaging of the left-sided tibialis anterior muscle with 3D motion capture of the ultrasound transducer’s position and orientation. Single transverse sweeping scans were performed from the proximal to distal (indicated by the horizontal dotted arrow) or distal to proximal direction at rest and during sustained isometric dorsiflexion contractions.

Prior to testing, participants performed five voluntary submaximal isometric dorsiflexion contractions (1-s hold, 1-s rest, ∼80% of maximum) to precondition the muscle–tendon unit (Maganaris, Baltzopoulos & Sargeant, 2002). Following this, two maximal isometric dorsiflexion contractions (3-s holds) were performed. In some cases, additional maximal contractions were performed, until there was no more than 5% difference in the maximum force attained.

Participants performed at least two ∼40-s isometric dorsiflexion contractions at 5, 10, 25 and 50% MVIC in a randomised order. Pilot testing revealed that the 50% MVIC level was the maximum that could be sustained without a drop in force over a 40-s period. Live visual feedback of the dorsiflexion force relative to time was provided via a monitor to enable participants to match the superimposed submaximal force levels. 3DUS scans of the entire TA muscle belly were performed during the contractions and at rest to assess muscle volume, muscle belly length, muscle CSA, muscle thickness and width, fascicle length and pennation angle, and central aponeurosis width and length. At least 1-min of rest was provided between contractions to minimise any possible fatigue effects. Participants were also asked and given more rest if required. Two scans were analysed separately from each condition to assess the intra-session repeatability of the measures.

Three-dimensional ultrasound

3DUS has been shown to be a valid and reliable technique for the in vivo measurement of human gastrocnemius muscle belly length and volume while muscles are relaxed (Barber, Barrett & Lichtwark, 2009). The 3DUS scan involved B-mode ultrasound imaging (SonixTouch, Ultrasonix, BC, Canada) with synchronous tracking of the ultrasound transducer’s position and orientation. A four-camera optical motion analysis system recording at 120 Hz (Optitrack, Natural Point, OR, USA) was used to track the 3D position of three spherical reflective markers (9.5 mm diameter) that were rigidly attached to the transducer (Fig. 2). Conventional B-mode ultrasound images of the TA muscle were acquired with a single transverse sweeping scan from the proximal to distal or distal to proximal end of the muscle belly and were subsequently transformed into the global coordinate system using Stradwin software (v5.0, Mechanical Engineering, Cambridge University, UK). Prior to scanning, the system was temporally and spatially calibrated using the single wall phantom calibration protocol available in Stradwin (Prager et al., 1998), as has been described elsewhere (Barber, Barrett & Lichtwark, 2009).

Transverse images of the soft tissues within the lower left leg were recorded at approximately 10–15 frames/s using a 60 mm linear transducer (L14-5W/60 Linear, Ultrasonix, BC, Canada) with a central frequency of 10 MHz and depth of 55 mm. A 2-cm thick echogenic ultrasound gel pad (Aquaflex, Parker Laboratories, NJ, USA) was positioned between the transducer and the skin to conform to the shape of the leg when light pressure was applied, ensuring that adequate contact was made across the skin so that transverse images of the TA muscle contained visible muscle borders. Echogenic gel was also applied over the skin to reduce friction during the sweeping scan. The total scan time was ∼35-s and the average distance between frames was ∼1 mm.

Force measurements

Net ankle force was determined using a custom-built rigid footplate with a force transducer attached close to the centre of the forefoot (Fig. 2). While the foot was constrained, contraction of the dorsiflexor muscles resulted in only small rotations at the ankle (<5° as determined previously during dorsiflexion MVICs using an identical experimental setup (Raiteri, Cresswell & Lichtwark, 2015)). Muscle and aponeurosis length changes and changes in the centre of pressure due to ankle rotation were therefore not accounted for as they were considered to be negligible. The force signal was amplified 500 times and sampled at 1 kHz.

Data analysis

Muscle CSA was manually segmented from sequential transverse images of the 3DUS scans at 5–15 mm intervals from approximately the level of the tibial tuberosity to ∼3 cm proximal to the level of the ankle malleoli. The same operator (BR) performed the segmentations across all scans in a randomised order on at least three separate occasions. The final segmentations were inspected in a reconstructed sagittal plane image re-slice of the muscle belly, where the plane could be translated in a medial-lateral direction relative to the line of action of the muscle. A very low resolution shape-based paradigm (Treece et al., 2000) was used to interpolate a surface through the segmented slices and algorithms available in Stradwin were used to turn the muscle cross-sections into a very highly smoothed 3D triangle-based model (Treece, Prager & Gee, 1999) to calculate muscle volume (Treece et al., 1999). The 3D mesh models were then exported for post-processing in Matlab (R2014b, MathWorks, MA, USA).

Muscle and central aponeurosis lengths were calculated as the straight-line distance between the most proximal and distal landmark locations in 3D space, relative to the laboratory coordinate frame. These landmarks were positioned in the approximate centre of the muscle or central aponeurosis in the first and last transverse slices where the structure was visible and were identified on at least two separate occasions. The landmark positions were confirmed in a reconstructed sagittal plane image re-slice of the muscle belly where the muscle and central aponeurosis end points were clear. The central aponeurosis was manually outlined along its width medially to laterally in the transverse images using landmarks that were spaced sequentially at 10–15 mm intervals along the length of the central aponeurosis. This was performed at least two times for each 3DUS scan and the order that scans were analysed was randomised. The 3D position of each central aponeurosis landmark within the laboratory reference frame was later exported for post-processing in Matlab.

A weighted principal component analysis (PCA) was performed in Matlab to determine the longitudinal axis of the muscle and its central aponeurosis separately. Before the PCA could be performed for the central aponeurosis, a triangular mesh was first reconstructed from the 3D position data based on a Delaunay algorithm (Image Processing Toolbox; Matlab, R2014b). The PCA involved subtracting the weighted mean (using the areas of triangles in the 3D triangular mesh as weights) from each of the data dimensions and calculating the eigenvectors and eigenvalues of the weighted covariance matrix. Then, a feature vector (x, y and z) was constructed from the eigenvectors that explained the greatest variance in the data in a descending order, so that the long axis corresponded to the x-axis, and each surface reconstruction (muscle or central aponeurosis) was transformed and rotated accordingly.

Figure 3 Reconstructions of the tibialis anterior muscle belly, central aponeurosis and muscle fascicles at rest.

(A) Individual example of a tibialis anterior (TA) muscle belly reconstruction (red) with a transverse cross-section (light blue) of the muscle reconstruction illustrating how muscle thickness and width measures were made. (B) Individual example of a TA central aponeurosis reconstruction (dark blue) with a birds-eye view illustrating how central aponeurosis width and length measures were made. Each image in (A) and (B) is represented in the coordinate system defined by the principal component analysis of the central aponeurosis. (C) Individual example of a reconstructed sagittal plane image re-slice of the TA muscle. The best sagittal plane image re-slice was determined visually as the image that displayed the clearest most continuous muscle fascicles. Three fascicles in the superficial and deep muscle compartments were analysed in the fascicle region outlined.

Following the PCA, the mesh vertices of the muscle reconstructions were down-sampled by a factor of 10 and a triangulated tight surface was returned based on a Crust algorithm (Giaccari, 2008). The mesh normals of both the muscle and central aponeurosis reconstructions were then unified and their feature vectors replicated to produce the coordinates of a 3D rectangular grid. From there, the muscle CSA and central aponeurosis width in the transverse plane (y–z plane) along the x-axis was calculated (Figs. 3A and 3B) at 1% increments of the total length from the centroid. This process essentially re-sliced the volume in the y–z plane of the PCA-axis system to ensure that there was no distortion due to the angle of the images that were taken relative to the muscle. To determine muscle thickness and width, the muscle surface reconstruction was transformed and rotated into the x, y and z-axes calculated by the PCA of the central aponeurosis from the respective contraction condition. Thickness and width of the PCA-transformed data in the y–z plane were calculated at the centre of the muscle slice at 1% increments of the total muscle belly length from the centroid. Muscle thickness was defined as the vertical distance of the muscle slice in the y-dimension and muscle width as the horizontal distance of the muscle slice in the z-dimension (Fig. 3A).

Fascicle length and pennation angle were measured at different regions within the muscle using a reconstructed sagittal plane image re-slice centered on the most proximal location of the central aponeurosis in 3D space. The final sagittal plane image re-slice for measuring fascicle lengths and pennation angles for one 3DUS scan was determined visually by rotating the image plane clockwise or anti-clockwise along the horizontal axis, until the image displayed the clearest and most continuous muscle fascicles, which was presumed to correspond to the plane of muscle fascicles (Narici et al., 1996). Three fascicles and their corresponding pennation angles (the angle of fascicle insertion into the central aponeurosis) were then measured in both the superficial and deep compartments of the muscle at proximal, middle and distal ends of the central aponeurosis (Fig. 3C). Because the fascicle curvature at rest and during contraction was small (e.g., Fig. 3C), which has been reported previously (Maganaris & Baltzopoulos, 1999), we felt that using a straight line to estimate fascicle length was an acceptable approximation of true fascicle length.

Statistics

Muscle segmentations were not performed on one subject due to difficulties in observing the lateral and deep borders of the TA muscle (n = 11 for muscle volume and muscle CSA measures). Central aponeurosis reconstructions from five subjects were excluded from analyses because of difficulties in either visualising the central aponeurosis or unreliable reconstructions because of high curvature and/or irregular shapes of the aponeurosis (n = 7). Muscle thickness and width measures were therefore performed on six individuals because one set of central aponeurosis reconstructions was performed on the participant with no corresponding muscle segmentations. Statistical analyses were performed using commercially available software (Prism 6, La Jolla, CA) and the level of significance was set at P ≤ 0.05. All data are presented as means ± standard deviation in the text and means ± standard error in the figures.

To determine the repeatability of the length and pennation angle measures within the same session, the intraclass correlation coefficient (ICC), the standard error of measurement (SEM) and the coefficient of variation (CV) were calculated. The coefficient of multiple correlation (CMC) was used to assess the intra-session repeatability of muscle CSA, thickness and width, and central aponeurosis width measures. Data were normally distributed according to Shapiro–Wilk normality tests. One-way repeated-measures ANOVAs were used to assess differences in muscle belly and central aponeurosis length changes, central aponeurosis width changes at 50% of aponeurosis length and muscle CSA, width and thickness changes at 50% of muscle belly length across the contraction conditions. Muscle fascicle length and pennation angle at rest in the superficial and deep muscle compartments were compared using a paired t-test and Wilcoxon signed-rank test, respectively. Differences in fascicle length and pennation angle changes between the muscle compartments across the contraction conditions were compared using two-way ANOVAs (contraction intensity × muscle compartment).

Table 1 Intra-session reliability of tibialis anterior muscle lengths, central aponeurosis lengths, fascicle lengths and pennation angles at rest and as a function of contraction intensity.

	Contraction intensity (% MVIC)	
	Rest	5	10	25	50	
Intra-class correlation coefficient2,1	
Muscle length	>0.99	
Central aponeurosis length	>0.99	
Fascicle lengths	>0.99	0.99	>0.99	>0.99	0.99	
Fascicle lengthd	0.98	0.99	>0.99	>0.99	0.99	
Pennation angles	0.82	0.89	0.59	0.88	0.49	
Pennation angled	0.94	0.99	0.95	0.93	0.96	
Standard error of measurement	
Muscle length (mm)	0.78	0.42	0.34	0.29	0.36	
Central aponeurosis length (mm)	0.26	0.41	0.43	0.41	0.36	
Fascicle lengths (mm)	0.18	0.10	0.14	0.17	0.18	
Fascicle lengthd (mm)	0.41	0.26	0.17	0.12	0.18	
Pennation angles (°)	0.47	0.26	0.63	0.38	0.66	
Pennation angled (°)	0.43	0.21	0.32	0.42	0.42	
Coefficient of variation (%)	
Muscle length	0.20	0.10	0.10	0.10	0.10	
Central aponeurosis length	0.20	0.20	0.30	0.20	0.20	
Fascicle lengths	0.30	0.30	0.20	0.30	0.30	
Fascicle lengthd	0.60	0.40	0.20	0.20	0.30	
Pennation angles	2.50	2.20	5.40	3.00	4.80	
Pennation angled	4.80	1.40	2.40	2.90	2.60	
Notes.

s superficial compartment

d deep compartment

MVIC Maximal Voluntary Isometric Contraction

Results

Reliability of measures

TA muscle segmentations for each subject resulted in TA muscle volume varying by only 2–7 mL (1–5%) across the ten segmentations performed at rest through to 50% MVIC. Muscle volume ranged from 142–284 mL across subjects. The intra-session reliability for muscle belly length, fascicle length and central aponeurosis length measures was excellent at rest and across the contraction conditions (Table 1: ICC = 0.98–>0.99; SEM = 0.10–0.78 mm; CV = 0.1–0.6%). Notably, the reliability for pennation angle measures was less, particularly measures made within the superficial compartment, as determined by the ICCs of 0.49–0.89. However, the absolute reliability of pennation angle measures was still high (Table 1), as evidenced by the small SEMs (≤0.66°), and the depressed ICCs were likely to be because of low between-subjects variability in pennation angles rather than increased trial-to-trial variability (Weir, 2005). CMCs for muscle CSA, muscle width and muscle thickness measures indicate that these measures were highly repeatable (range = 0.98–>0.99). CMCs for central aponeurosis widths were slightly lower, but within an acceptable range (range = 0.89–0.99). These methods were therefore deemed reliable within the same session and were used in additional analyses regarding muscle and aponeurosis shape changes as a function of contraction intensity.

Figure 4 Tibialis anterior muscle length changes during isometric dorsiflexion contractions as a function of contraction intensity (% of maximal voluntary isometric contraction).

Negative values indicate shortening. Data are presented as mean ± standard error.

Figure 5 Tibialis anterior muscle deformations perpendicular to its line of action as a function of contraction intensity.

(A) Muscle cross-sectional area (CSA), (B) muscle width and (C) muscle thickness changes of the tibialis anterior at 25–75% of muscle length during isometric dorsiflexion contractions at 5%, 10%, 25% and 50% of maximal voluntary isometric contraction. Data are presented as mean ± standard error.

Table 2 Changes in tibialis anterior muscle parameters perpendicular to its line of action at 50% of muscle length as a function of contraction intensity.

Muscle parameter	Contraction intensity (% MVIC)	P value	
	5	10	25	50		
Δ Muscle CSA (mm2)	28.3 ± 21.6	36.5 ± 26.3	50.3 ± 29.7	63.3 ± 47.7	<0.01	
Δ Muscle thickness (mm)	1.2 ± 1.0	1.4 ± 0.6	2.0 ± 0.6	2.5 ± 0.8	<0.01	
Δ Muscle width (mm)	−0.7 ± 1.9	−0.8 ± 1.7	0.3 ± 1.4	0.4 ± 1.7	0.35	
Notes.

MVIC Maximal Voluntary Isometric Contraction

CSA Cross-Sectional Area

Muscle shape changes during contraction

TA muscle belly length decreased as contraction intensity increased (Fig. 4) and there was a significant main effect of contraction intensity on muscle belly length (P < 0.01). Because muscle volume remained relatively constant across contraction conditions as the muscle belly shortened, muscle CSA increased with contraction effort. Peak increases in CSA centred around the middle of the muscle belly (Fig. 5A). At 50% of muscle belly length, there was a significant main effect of contraction condition on muscle CSA (P < 0.01; Table 2). These increases in muscle CSA were mainly driven by increases in muscle thickness rather than muscle width (Figs. 5B and 5C). A significant main effect of contraction intensity on muscle thickness was observed at 50% of muscle belly length (P < 0.01; Table 2), however the same effect was not found on muscle width (P = 0.35; Table 2).

Fascicle length and pennation angle changes during contraction

TA muscle fascicle length decreased as contraction intensity increased in both the superficial and deep muscle compartments (Fig. 6A), while pennation angle increased (Fig. 6B). The decrease in fascicle length, as a function of contraction intensity, was curvilinear. At rest, the superficial muscle compartment had significantly shorter muscle fascicles than the deep muscle compartment (68.6 ± 2.2 mm and 72.4 ± 2.5 mm, respectively; P < 0.01), however pennation angle was not significantly different (10.9 ± 0.57° and 11.1 ± 1.5°, respectively; P = 0.27). A comparison of fascicle length changes in the superficial and deep muscle compartments across the contraction conditions revealed a significant main effect of contraction intensity (P < 0.01), but no main effect of muscle compartment (P = 0.15) and no significant interaction (P = 0.98). The same comparison for pennation angle changes showed significant main effects of both contraction intensity (P < 0.01) and muscle compartment (P < 0.01), but no significant interaction (P = 0.94).

Figure 6 Tibialis anterior muscle architecture changes during isometric dorsiflexion contractions as a function of contraction intensity.

(A) Muscle fascicle length and (B) pennation angle changes of the tibialis anterior in the superficial and deep compartments at 5%, 10%, 25% and 50% of maximal voluntary isometric contraction. Negative values in (A) indicate shortening. Data are presented as mean ± standard error.

Central aponeurosis deformations during contraction

There was a significant main effect of contraction intensity on both central aponeurosis width and length changes (P < 0.01). Mean central aponeurosis width decreased slightly at 5% MVIC, although this was not significantly different from zero. Increases in aponeurosis width were then apparent from 5 to 50% MVIC (Figs. 7A and 7B). Central aponeurosis length increased in all of the contraction conditions (P < 0.01; Fig. 7C). The strains in the transverse direction at 50% of central aponeurosis length were greater than the strains in the longitudinal direction at the same contraction intensity (Figs. 7B and 7C). This was primarily due to differences in the resting length rather than the absolute stretch of the tissue in each direction (Table 3).

Figure 7 Tibialis anterior central aponeurosis deformations during isometric dorsiflexion contractions as a function of contraction intensity.

(A) Central aponeurosis (CA) width changes of the tibialis anterior at 10–90% of central aponeurosis length, (B) central aponeurosis width changes at 50% of central aponeurosis length and (C) central aponeurosis length changes at 5%, 10%, 25% and 50% of maximal voluntary isometric contraction. Data are presented as mean ± standard error.

Table 3 Tibialis anterior central aponeurosis width and length changes as a function of contraction intensity.

Central aponeurosis parameter	Contraction intensity (% MVIC)	P value	
	5	10	25	50		
Δ Central aponeurosis width (mm)	−0.8 ± 3.8	1.0 ± 3.5	4.4 ± 3.6	6.2 ± 3.3	<0.01	
Δ Central aponeurosis length (mm)	1.6 ± 0.5	3.1 ± 0.8	4.3 ± 1.1	6.0 ± 1.5	<0.01	
Notes.

MVIC Maximal Voluntary Isometric Contraction

Discussion

The results of this study provide the first evidence of how the human TA muscle bulges and how its central aponeurosis deforms in vivo with increasing contraction intensity during isometric dorsiflexion. The 3DUS technique was suitably reliable and sensitive to make the measures required for assessing muscle shape changes during isometric contraction. The TA muscle belly remained isovolumetric from rest through to 50% MVIC and progressively shortened along its line of action as contraction intensity increased, which caused the muscle belly to bulge centrally, as expected. This bulging orthogonal to the muscle’s line of action presumably occurred because of the transverse strain present in the muscle fibres as they shortened (Wakeling & Randhawa, 2014) and were constrained to maintain a constant volume (Huxley, 1953; Elliott, Lowy & Worthington, 1963). Fascicle shortening was accompanied by fascicle rotation and increases in muscle thickness at the centre of the muscle. Muscle fascicle shortening and presumably radial expansion of muscle fibres (Wakeling & Randhawa, 2014) caused the central aponeurosis to strain in both longitudinal and transverse directions in a curvilinear fashion with increasing contraction intensity within the range tested here.

The increase in muscle belly shortening observed at higher contraction intensities is not surprising given that muscle fascicle lengths also progressively decreased with increasing contraction intensity at roughly equivalent magnitudes. This indicates that muscle fascicle length changes were likely to be relatively uniform throughout the muscle. Because muscle fibre volume remains relatively constant during contraction (Huxley, 1953; Elliott, Lowy & Worthington, 1963), active fibre shortening must be accompanied by radial expansion of muscle fibres, and these multidirectional forces generated within muscle drive complex changes in muscle shape (presumably through changes in intramuscular pressures). We found that muscle shortening was accompanied by significant increases in muscle CSA, predominantly around the middle of the muscle. This central muscle bulging was likely to be because the muscle fibres shortened the most in this region (Rahemi, Nigam & Wakeling, 2014) or because the physiological CSA was greatest in the middle of the muscle and therefore the transverse forces applied to the muscle were likely to be the highest here.

Increases in muscle CSA were predominantly driven by increases in muscle thickness, rather than muscle width. This finding contradicts results reported by Maganaris and colleagues (1999), who found that TA muscle thickness remained constant during maximal voluntary contraction, and is likely to be because of the different contraction intensities used. The increases in thickness with increasing contraction intensity are perhaps expected when pennation angle increases during contraction, however this also depends on how the muscle bulges (Randhawa & Wakeling, 2015). Data from Azizi and colleagues (2008) showed that muscle thickness significantly decreased with the force of contraction in the lateral gastrocnemius muscle of turkeys, which suggests that lower contraction intensities may favor fast contractions (i.e., the velocity of muscle belly shortening would exceed the velocity of muscle fibre shortening) over forceful contractions. The reason that muscle thickness increases are favoured at lower forces is unknown, but it may be because of increased stiffness of the connective tissue elements in the direction of muscle width compared with muscle thickness (e.g., the relative thickness of the perimysium along muscle fascicles in the direction of width versus thickness may be greater (Sharafi & Blemker, 2010)). Alternatively, aponeurotic sheets above and below the muscle fibres may permit muscle thickness expansion because they have a low transverse stiffness and strain more in this direction (potentially due to cross-fibred collagen arrays (Kannus, 2000)) relative to the fascia surrounding the medial and lateral aspects of the TA.

The muscle thickness increases we observed might also vary from the Maganaris & Baltzopoulos (1999) study because of the different methods used to calculate thickness. Our method of re-slicing the TA muscle in the y–z plane along the longitudinal axis of its central aponeurosis (as determined by a PCA) enables accurate measures of thickness that might not be possible using a fixed image plane with conventional 2DUS, where it is clear that measurement error is associated with transducer alignment (Bolsterlee, Gandevia & Herbert, 2016). The fixation of the transducer to the leg in 2DUS might also cause greater muscle compression compared to 3DUS (Wakeling, Jackman & Namburete, 2013), which may prevent the muscle from increasing in thickness.

The observation that fascicle rotation increased with fascicle shortening was expected (Narici et al., 1996) and is in agreement with the thickness increases observed in the centre of the muscle (this is where fascicle lengths and pennation angles were primarily measured). Although we did not measure fascicle lengths and pennation angles in the proximal unipennate portion of the muscle or the most distal end, the mean pennation angle of 11° we observed at rest is similar to the mean values reported by Maganaris and colleagues (1999), who used 2DUS to image the TA mid-belly, and Hiblar and colleagues (2003), who used 3DUS to measure pennation angles throughout the entire TA muscle belly. The investigation by Hiblar et al. (2003) found that pennation angles in the proximal part of the muscle (15°) were greater than pennation angles at the distal end (7°) and that the angle of insertion decreased in the medio-lateral plane towards the boundary of the muscle (where fascicles were curved slightly). Because larger pennation angles have been associated with greater amounts of fascicle rotation during contraction (Shin et al., 2009), this may allow the longer proximal fascicles (evidenced by greater separation between aponeuroses) to shorten at similar magnitudes and velocities as the distal fascicles during contraction, extending the range that the TA muscle is mechanically efficient.

Fascicle shortening was shown to stretch the central aponeurosis of the TA in a curvilinear manner with respect to contraction intensity. This result has been documented previously at contraction intensities above 10% MVIC (Maganaris, Kawakami & Fukunaga, 2001; Tilp, Steib & Herzog, 2012). However, at contraction intensities of 10% MVIC, Tilp and colleagues (2012) found that the central aponeurosis length decreased, which is not in line with what we observed at the same contraction intensity. We believe that this difference may be due to difficulties in accurately determining points on the aponeurosis along the longitudinal axis of the muscle when making 2DUS measurements. This may be particularly pertinent at low muscle forces, where there is heterogeneity in force distribution along the aponeurosis (Zuurbier et al., 1994). By using exact end-points of the aponeurosis, we were able to determine a global strain more accurately than with a 2D approach. Our findings suggest that the central aponeurosis acts in-series with the contractile element, where it lengthens in proportion to muscle fascicle shortening, at low and moderate forces. Given that the aponeurosis is an elastic material, it therefore likely contributes a significant amount to the storage and return of elastic energy.

The curvilinear relationship that we observed between contraction intensity and central aponeurosis strain suggests that the stiffness of the central aponeurosis increased with force, which is in-line with the known properties of tendinous tissue. A curvilinear relationship between longitudinal aponeurosis stretch and muscle activation has previously been observed in the TA muscle (Maganaris & Paul, 2000). The relatively large amount of aponeurosis elongation at 5% and 10% MVIC may be attributed to the initial extension allowed by the crimp of collagen fibrils (Diamant et al., 1972), which is commonly known as the ‘toe-region’ (Ker, 1981). At higher contraction intensities (25% and 50% MVIC in our study), the stretch of collagen fibrils in the aponeurosis presumably gave rise to linear increases in aponeurosis length outside of the so-called ‘toe-region’. A greater stiffness of the aponeurosis at the higher contraction intensities reduced the change in muscle fascicle length and pennation angle (i.e., the relationship between fascicle length/pennation angle change and contraction level was also curvilinear) for a given change in force output (intensity).

Central aponeurosis strains were greater in the transverse direction (14.4%) compared with the longitudinal direction (4%) and we believe that this is a function of the material properties of the aponeurosis and a requirement for radial expansion of muscle fibres. The biaxial stretching of the aponeurosis was probably related to increases in muscle fibre CSA above and below the aponeurosis (Scott & Loeb, 1995) as fibres maintained a constant volume during shortening (Elliott, Lowy & Worthington, 1963). The increase in fibre CSA was presumably accounted for by increases in fibre diameter in both the transverse (which was not significant at 5% MVIC) and sagittal planes, and the stretching of the aponeurosis along these dimensions probably reduced its thickness in the frontal plane.

The transverse central aponeurosis strains we measured are in line with previous estimates based on 2DUS imaging (Maganaris, Kawakami & Fukunaga, 2001; Muraoka et al., 2003). Relative to the longitudinal strains, the transverse aponeurosis strains may have been greater because of underestimations of central aponeurosis width or because of differences in aponeurosis compliance in the transverse and longitudinal directions. This latter explanation is supported by findings from Azizi and colleagues (2009), who observed a five times smaller transverse elastic modulus compared with the longitudinal elastic modulus of the superficial lateral gastrocnemius aponeurosis of turkeys. The greater stiffness of the aponeurosis in the longitudinal direction was attributed to the primarily longitudinal arrangement of collagen fascicles along the muscle’s line of action, while the structures responsible for the low transverse aponeurosis modulus were not confirmed (Azizi, Halenda & Roberts, 2009). Interestingly, transverse strains of the aponeurosis were later shown to modulate the longitudinal stiffness of this aponeurosis (Azizi & Roberts, 2009). We similarly observed increases in the longitudinal stiffness of the TA central aponeurosis (curvilinear strain vs. intensity relationship), however it is unclear whether this relationship was due to the normal longitudinal properties of tendinous tissue (discussed earlier) or due to stiffness modulation from transverse aponeourosis strains.

It is certainly worth investigating if factors other than muscle force (such as muscle length and activation) can affect the magnitudes of fascicle shortening and subsequent radial expansion of muscle fibres, which in turn may alter transverse aponeurosis strains (Muraoka et al., 2003) and the longitudinal stiffness of the aponeurosis. Implementing 3DUS to measure aponeurosis strains at different muscle lengths is feasible, provided that an experienced ultrasound operator conducts the scans on a muscle with clearly identifiable borders and a well-defined aponeurosis that can be captured with one sweep of the ultrasound transducer. Ideally, joint rotation should be minimised so that aponeurosis length changes are not overestimated due to passive length changes during contraction and the transducer’s orientation about its horizontal and longitudinal axes should be maintained throughout the scan so that the transducer is perpendicular to the long axis of the muscle for as much of the scan as possible. This is because deviations in transducer pitch or roll may result in transverse images overlapping and poor image quality that prevents the muscle or aponeurosis from being segmented accurately (we made our statistical comparisons relative to 50% of muscle and aponeurosis length because we were confident that the transducer was perpendicular to the muscle and aponeurosis here). The use of an ultrasound gel pad between the transducer and the skin would also be helpful for minimising the amount of muscle compression, which in our study we believe was small (we estimate that muscle thickness may have been reduced by ∼0.5 mm at rest due to transducer pressure and to a lesser extent during the isometric contractions because of the higher TA intramuscular pressures (Aratow et al., 1993; Styf et al., 1995)) and therefore unlikely to have altered the main effect we observed of contraction intensity on muscle thickness.

Conclusions

Our results show that 3DUS is a viable and reliable method for quantifying multidirectional muscle and aponeurosis deformations during contractions within the same session and that contraction intensity (i.e., force output) is an important factor for determining the magnitude of TA muscle bulging and shortening, as well as the amount that the central aponeurosis strains in both the longitudinal and transverse directions. It appears that contracting muscle fibres do work in directions along and orthogonal to the muscle’s line of action (evidenced by increases in muscle belly shortening and muscle CSA). We have shown that central aponeurosis length and width appear to be a function of muscle fascicle shortening (and presumably subsequent transverse expansion of muscle fibres), and that there are significant transverse strains, which are likely to indicate transverse loading. How these aponeurosis deformations influence the storage and return of elastic strain energy and if factors other than muscle force change the elastic mechanical behaviour of the aponeurosis requires further investigation.

Supplemental Information

Supplemental Information 1 Absolute tibialis anterior central aponeurosis lengths determined from each ultrasound scan for each participant

Click here for additional data file.

Supplemental Information 2 Absolute superficial compartment tibialis anterior fascicle lengths and pennation angles determined from each ultrasound scan for each participant

Click here for additional data file.

Supplemental Information 3 Absolute deep compartment tibialis anterior fascicle lengths and pennation angles determined from each ultrasound scan for each participant

Click here for additional data file.

Supplemental Information 4 Absolute tibialis anterior muscle volumes determined from each ultrasound scan for each participant

Click here for additional data file.

Supplemental Information 5 Average tibialis anterior muscle lengths (mm) determined for each participant

Click here for additional data file.

Supplemental Information 6 Average tibialis anterior central aponeurosis lengths (mm) determined for each participant

Click here for additional data file.

Supplemental Information 7 Average tibialis anterior muscle volumes (mL) determined for each participant

Click here for additional data file.

Supplemental Information 8 Absolute tibialis anterior muscle lengths determined from each ultrasound scan for each participant

Click here for additional data file.

Supplemental Information 9 Tibialis anterior muscle and central aponeurosis dimensions perpendicular to the line of action of the muscle for each participant

(A) Muscle cross-sectional area (CSA), (B) muscle thickness and (C) muscle width from 25–75% of muscle length and (D) central aponeurosis width from 10–90% of central aponeurosis length for each participant at rest and across the contraction conditions.

Click here for additional data file.

Additional Information and Declarations

Competing Interests

Author Contributions

Human Ethics

Data Availability

The authors declare there are no competing interests.

Brent J. Raiteri conceived and designed the experiments, performed the experiments, analyzed the data, contributed reagents/materials/analysis tools, wrote the paper, prepared figures and/or tables.

Andrew G. Cresswell and Glen A. Lichtwark conceived and designed the experiments, contributed reagents/materials/analysis tools, reviewed drafts of the paper.

The following information was supplied relating to ethical approvals (i.e., approving body and any reference numbers):

The University of Queensland Human Research Ethics Committee; approval number: HMS14/0704.

The following information was supplied regarding data availability:

The raw data has been supplied as a Supplemental File.

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
