# Peer review of "Three-dimensional geometrical changes of the human tibialis anterior muscle and its central aponeurosis measured with three-dimensional ultrasound during isometric contractions"

_PeerJ, doi:10.7717/peerj.2260_

## Round 0.1 · original submission · Minor Revisions

I have received two reviews of your paper. Both reviewers consider that your manuscript provides interesting insights in a research developing area. They also consider your data sound and reliable, with appropriate interpretations. Yet, there are a number of points that need your further consideration. I would like you to add a cartoon type figure to define the geometrical terms, and a picture of the experimental setup. Please take care of the missing labels on the figures.

I would like to see a revised version of your manuscript that takes a point by point account of the comments of the reviewers.

Reviewer 1 ·

Basic reporting

The manuscript provides interesting insights into a developing area of research: 3D deformation of muscles during contractions. The reliability of a relatively new 3D ultrasound method is tested and applied to study changes in the 3D shape of the tibialis anterior. Overall the manuscript is well structured and written and describes a substantial amount of work. The figures and tables are well designed.
I have two suggestions regarding reporting:
Line 20-21: ‘difficulties with performing non-invasive procedures’. Please be more specific in what these difficulties are. Please also rephrase in line 69.
Line 70: It is mentioned that only one study has determined aponeurosis length and width changes, but in the next lines two studies are described (Maganaris 2001 and Tilp 2012). Please clarify.

Experimental design

The experiments were carefully designed and are in general clearly described. The data analysis, which provides a substantial amount of the work, is also well described. However, I would like the authors to clarify the following points:
Line 112: How was ankle angle measured and defined?
Line 206-207: What is meant by ‘grid vectors’ and why do they need to be calculated? Can’t the width and the depth be calculated directly from the y- and the z-dimension of the PCA-transformed data? Please clarify this step.
Line 213: What is meant by horizontal and vertical distance? Are these the y- and z-dimension of the PCA transformed data?
Line 221-222: How repeatable was this procedure for finding the angle at which fascicles appeared clearest?

Validity of the findings

Although care is taken to produce reliable results (especially by doing all measurements twice to test the reliability), there are a few things that I would like the authors to comment on:
Line 155: Light pressure is applied on the transducers to maintain contact with the gel-pad. Even if constant pressure was applied during the contractions, the difference in intramuscular pressure between the conditions would result in differences in compression of the muscle by the transducer. Could you indicate whether this could have significantly affected the measurements (presumably mostly the muscle depth measurements when compared between rest and active contractions)?
Line 223-226: I assume that the pennation angle is calculated as the angle between the fascicle and the tangent line to the aponeurosis in the 2D resliced US plane. Using the 3D surface models the 3D pennation angle (i.e. the angle of the fascicle to the tangent plane of the aponeurosis) could be calculated as well. Why is the 2D and not the 3D pennation angle used? How does the choice of using the 2D pennation angle influence the results?
Line 234: The central aponeuroses of 5 out 12 subjects were excluded. Given that this is almost half of the data, it should be made more clear what criteria were used to exclude these subjects. Could you speculate on why the aponeuroses of some subjects were better visible than others? This should be discussed in the Discussion as a limitation of the technique.
Line 248-251: Why were only values at 50% of muscle belly length used in the statistical analysis? Is there a reason why just these values were used? If this was a rather arbitrary choice made after inspecting the data this should be made clear.

Additional comments

My only further comment is that it would be a good addition to the manuscript if the authors could elaborate a bit more on the limitations and applicability of the 3D ultrasound technique.

Reviewer 2 ·

Basic reporting

Good. Some clarifications on figure labels would help. See below.

Experimental design

Sound with very detailed description.

Validity of the findings

Good.

Additional comments

Manuscript Number: PeerJ #10333
Three dimensional geometrical changes of the human TA muscle and its central aponeurosis measured with #D ultrasound during isometric contractions

SUMMARY OF WORK:
The purpose of this study was to employ a novel 3D ultrasound technique to provide an updated view of how human muscle and elastic tissue geometry changes as active force increases during isometric contractions. The muscle studied here was the tibialis anterior- with superficial and deep compartments around a central aponeurosis.

The results demonstrated good reliability in the technique to recover measures of length, area and volume within a given experimental test session. Furthermore, the authors nicely demonstrated that as contractile force increases, fascicles shorten, rotate and bulge while aponeurosis is deformed in concert.

In its current form, the manuscript flows logically, is very clearly written and provides novel data demonstrating an exciting approach and some important fundamental geometrical relationships in detail not seen before. I have a number of minor suggestions for improving the clarity and scope of the work.
* * *
COMMENTS:

Major Compulsory Revisions (which the author must respond to before a decision on publication can be reached)
1. n/a

-Minor Essential Revisions (such as missing labels on figures, or the wrong use of a term, which the author can be trusted to correct)

1. Introduction –general- I found it difficult to follow aspects of the introduction because of incomplete knowledge of the anatomy being discussed. It would help to refer to a cartoon type figure (maybe Fig. 1 modified a bit) that carefully defines the geometrical terms (volume, length, width, CSA, pennation angle of muscle etc.) more clearly up front.

2. Introduction –lines 69-71, 84-85 I seem to recall a study by Farris et al. that used a similar 3D technique as is employed here to study AT deformation in more than 1 plane. This might be worth acknowledging in the setup.

J Exp Biol. 2013 Feb 15;216(Pt 4):594-600. doi: 10.1242/jeb.077131. Epub 2012 Nov 1.
Differential strain patterns of the human Achilles tendon determined in vivo with freehand three-dimensional ultrasound imaging.
Farris DJ1, Trewartha G, McGuigan MP, Lichtwark GA.

3. Methods –line 109+
A picture of the exp. setup would be nice to help investigators reproduce the setup. Especially the footplate and load cell setup.

4. Methods –line 161+
Can you address concerns that COP on the footplate may have been dynamically changing and therefore causing a changing mechanical advantage of the muscle force to foot plate force thereby causing inaccuracies in the force levels applied via biofeedback.

5. Methods –line 187+ Can you provide some more detail on how ‘landmarks’ were chosen in the images.

6. Results –line 301+ How can both length and width of the aponeurosis increase above 5%MVIC? Is apo thickness decreasing? Please reconcile.

7. Discussion –line 349+ Could material properties of the aponeurosis (not just muscle connective tissues) also help explain bias for muscle to change thickness rather than width?

7. Discussion –general- Based on your data can you speculate on whether some portion of muscle work is ‘wasted’ in directions that are not locomotion relevant? If so, how much? And why? What is the trade-off?

FIGURE 1. Please add some detail on this figure to clarify proximal , distal, anterior posterior, lateral, medial directions with respect to the limb. Define pennation angle and fascicle length.

FIGURES. In general, it is difficult to tell whether the definition of length increasing from 0% to 100% goes from distal to proximal or vice versa. Please label this in Fig 1, 3, 5A for example.

It seems some data is repeated in Tables and Figures. This is fine, but maybe adding stats to the Tables would make them more useful as an addition to the Figures.

---

## Round 0.2 · accepted · Accept

I thank the authors for the great work they did in considering the reviewer's suggestions. I strongly recommend to set this as a public review. The interchange between both, authors and reviewers, would be a great help to other researchers in the field.